# The Biomechanical Behavior of Distal Foot Joints in Patients with Isolated, End-Stage Tibiotalar Osteoarthritis Is Not Altered Following Tibiotalar Fusion

**DOI:** 10.3390/jcm9082594

**Published:** 2020-08-11

**Authors:** Maarten Eerdekens, Kevin Deschamps, Sander Wuite, Giovanni Matricali

**Affiliations:** 1Department of Rehabilitation Sciences, Musculoskeletal Rehabilitation Research Group, KU Leuven, 3001 Heverlee, Belgium; 2Clinical Motion Analysis Laboratorium (CMAL), UZ Leuven, 3210 Pellenberg, Belgium; kevin.deschamps@kuleuven.be; 3Musculoskeletal Rehabilitation Research Group, Department of Rehabilitation Sciences, KU Leuven, 8000 Campus Brugge, Belgium; 4Department of Orthopedics, UZ Leuven, 3000 Leuven, Belgium; sander.wuite@uzleuven.be (S.W.); giovanni.matricali@uzleuven.be (G.M.); 5Institute for Orthopaedic Research & Training, KU Leuven, 3000 Leuven, Belgium; 6Department of development and regeneration, KU Leuven, 3000 Leuven, Belgium

**Keywords:** ankle, arthrodesis, foot, biomechanical phenomena, gait analysis

## Abstract

Ankle arthrodesis is considered to be an optimal treatment strategy to relieve pain during walking in patients with isolated, end-stage tibiotalar osteoarthritis. The aim of this study was to investigate the post-operative effect of an arthrodesis on the ankle and foot joint biomechanics. We included both patients (*n* = 10) and healthy reference data (*n* = 17). A multi-segment foot model was used to measure the kinematics and kinetics of the ankle, Chopart, Lisfranc, and first metatarsophalangeal joints during a three-dimensional (3D) gait analysis. These data, together with patient reported outcome measures, were collected at baseline (pre-operative) and one year post-operatively. Patients experienced a decrease in pain and an increase in general well-being after surgery. Compared to the baseline measurements, patients only demonstrated a significant average post-operative increase of 0.22 W/kg of power absorption in the ankle joint. No other significant differences were observed between baseline and post-operative measurements. Current findings suggest that the biomechanical behavior of distal foot joints is not altered one year after fusion. The pain relief achieved by the arthrodesis improved the loading patterns during walking. Clinical significance of this study dictates that patients do not have to fear a loss in biomechanical functionality after an ankle arthrodesis.

## 1. Introduction

Tibiotalar osteoarthritis (OA) is a disabling disorder characterized by progressive joint degeneration, pain, and functional impairments causing difficulties during daily life activities, such as walking [1,2]. Ankle arthrodesis (fusion of tibiotalar joint) is considered the gold-standard operative treatment strategy in patients with end-stage ankle OA, as it provides significant pain relief and long-term survival [3,4]. It has, however, been hypothesized that fusing the tibiotalar joint potentially leads to altered mechanical loads and maladaptive motion patterns at other foot joints [5,6], which would lead to adjacent joint degeneration, and consequently to further functional impairments [7]. Several authors have investigated the biomechanical effects of an ankle arthrodesis during walking. Flavin et al. [6] and Hahn et al. [8] have shown that patients with isolated tibiotalar OA demonstrated only a minimal reduction of ankle joint range of motion (RoM) [6,8]. Brodsky et al. found an increased ankle joint RoM after a tibiotalar fusion [9]. In their rationale, the authors stated that regardless of the articulations across which sagittal plane movement occurs post-operatively, some patients experience the persistence of mobility rather than the expected dramatic increase in stiffness. However, these contradictory findings might be attributed to methodological discrepancies between both studies, as the latter used a one-segment kinematic foot model, which leads to opposite results on ankle joint kinematics during gait [10].

Clinical experience dictates that patients with end-stage tibiotalar OA, with plans for an ankle arthrodesis, often fear a post-operative loss of functionality due to stiffness of the ankle and foot joints. Therefore, additional research during post-operative follow-up to monitor potential compensatory mechanisms in adjacent joints is mandatory. It is, for example, believed that the disappearance of pain in the tibiotalar joint might lead to increased load in distal foot joints, due to gait adaptations [11]. Gait analysis research should therefore perhaps not only focus on the expected gait biomechanical changes, but also investigate the potential function loss and biomechanical alterations after an ankle arthrodesis in the adjacent (foot) joints.

Indeed, a recent systematical review underlined the need for enhancing our understanding of the functional compensatory adaptations of adjacent joints—especially neighboring foot joints in patients with tibiotalar OA [11]. In the past, this has been difficult, due to technical limitations in three-dimensional (3D) gait analysis [12]. Recent efforts led to the development of novel, 3D, multi-segment kinetic foot models that have the possibility to quantify the biomechanical functioning of the complete ankle and foot complex [13,14]. These can furthermore provide us with an enhanced understanding of the functional adaptations of adjacent foot joints in patients with end-stage tibiotalar OA after surgical fusion.

The aim of this study was therefore to compare pre- and one year post-operative gait biomechanics of the foot and ankle complex during walking in patients with isolated, end-stage tibiotalar OA receiving an ankle arthrodesis. We only selected patients with isolated tibiotalar OA, to ensure that the distal foot joints were free of OA and were thus potentially able to adopt another mechanical behavior. Our hypothesis was that the kinetic behavior of the adjacent foot joints would be less impaired post-operatively, since pain-mediated motion patterns would be elevated by the tibiotalar joint fusion.

## 2. Material and Methods

### 2.1. Participants

From a prospectively collected database generated by our research group, we selected pre-and one year post-operative gait analysis and questionnaire data of 10 patients with isolated end-stage ankle OA. For reference, we also included gait analysis data of 17 asymptomatic control subjects (Table 1). All patients were diagnosed with isolated end-stage ankle OA, which was defined as OA of only the tibiotalar joint at one side, by two senior orthopedic surgeons (G.M. and S.W.) of the Foot and Ankle Unit of the University Hospitals of Leuven. Patients were included whenever a Kellgren OA score of 4 was unilaterally confirmed in the tibiotalar joint using X-ray imaging, and if former conservative treatments had failed. Exclusion criteria for both the patient and control group were (I) not being able to walk at least 100 m without walking aids; (II) any trauma or medical condition (other than ankle OA for patients) that could affect normal gait, such as systemic, neurological, acute, or chronic diseases affecting lower limb joints; and (III) being younger than 18 year of age. Patients were furthermore excluded whenever a Kellgren OA score of >2 was observed for the subtalar, talonavicular, or calcaneocuboid joint. The study was conducted in accordance with the Declaration of Helsinki, and was approved by the local ethical committee (S55070; ML9038). Prior to participating, all subjects read and signed an informed consent.

### 2.2. Gait Analysis

Pre-operative gait analyses were planned within two weeks before surgery, and post-operative analyses were planned 1 year after surgery. Measurements began with the placement of passive retro-reflective markers on anatomical landmarks of both feet, according to the Rizzoli Foot Model [15]. We selected this foot model because of its good repeatability in the assessment of multi-segment foot biomechanics (Mean inter-trial Coefficients of Multiple Correlation (CMCs) of >0.820) [16,17]. The placement and skin-motion artifact issues of the retro-reflective skin markers have been investigated previously with a test–retest study design within a similar patient population, with results suggesting not only relative 3D rotations and planar angles that can be measured consistently in patients, but also a number of absolute parameters that can be consistently measured, serving as basis for the decision-making process [17]. All participants performed at least four representative walking trials at a self-selected, comfortable speed in a clinical motion analysis laboratory consisting of a 3D motion analysis system (Vicon Motion System Ltd., Oxford Metrics, UK) and a plantar pressure platform (FootscanTM sensors, 2.8 sensors per cm2, RSscan International, Olen, Belgium) placed on top of a force platform (Advanced Mechanical Technology Inc., Watertown, MA, USA). The measurement accuracy of this motion analysis system was reported to be excellent, with a dynamic resolution and accuracy of 0.1 mm ± 0.89 mm and 99.4% [18], with the smallest range of motion (RoM) being seven times larger than the mean measurement error [19].

The kinetic computation combined the marker position, ground reaction force, and plantar pressure data. The center of pressure and resultant ground reaction force were distributed over four segments of the Rizzoli Foot Model (rearfoot, midfoot, forefoot, and hallux segment) using a proportionality scheme [14]. This was furthermore validated in a similar patient population [14]. Inertial parameter calculations of foot segments were based on the mass of the segments and on their geometric solids, whereas the mass of the foot was distributed at 30/30/30/10% for the rearfoot/midfoot/forefoot/hallux respectively. The inter-segment joint angles were defined as follows: ankle joint (between the shank and rearfoot segment), Chopart joint (between the rearfoot and midfoot segment), Lisfranc joint (between the midfoot and forefoot segment) and the first metatarsophalangeal joint (MTP 1; between the forefoot and hallux segment). An inverse dynamic analysis program written in Matlab (MathWorks, Natick, MA, USA) computed joint moments and powers starting from the hallux segment and progressing proximally using Newton–Euler equations, similar to a recent methodology published by Deschamps et al. [13] and Eerdekens et al. [14]. All one-dimensional data (waveforms) were normalized to 100% of the stance phase, and the mean was calculated based on the four recorded trials per subject.

The following kinematic outcome variables were withheld from the gait analyses for the ankle, Chopart, Lisfranc, and MTP 1 joints: kinematic waveform data (joint angles) in all three planes (sagittal, frontal, and transverse plane). From these kinematic waveforms, we further analyzed the active RoM experienced during the stance phase of walking, which was calculated by subtracting the minimum from the maximum value. We also withheld the following kinetic outcome variables: kinetic waveform data (internal joint moment (Nm/kg), joint angular velocity (°/s), and joint power (W/kg) in the sagittal plane). For the ankle joint, we also calculated the biomechanical load (bodyweight or BW), loading rate (BW/s), and time-to-peak (s).

### 2.3. Patient-Reported Outcomes

Patients were asked to fill out the Foot Function Index (FFI) and Short Form-36 (SF-36), and to score their pain using a Visual Analog Scale (VAS) at baseline (pre-operative) and one year post-operation. The FFI quantifies foot health and pain and the foot health-related quality of life. Scores are interpreted from 0 to 100, and a higher score indicates a poor outcome [20]. The SF-36 questionnaire scores eight health concepts concerning physical and mental well-being, and a higher score indicates a good outcome [21]. Both questionnaires were chosen for their validity, reliability, and responsiveness in this type of population [22].

### 2.4. Statistical Analyses

Waveform data were included for visual comparison between pre-and post-operative gait patterns. Peak and discrete variables were statistically compared using a paired *t*-test (α = 0.05). Cohen’s D effect size was investigated by calculating the mean difference between pre- and post-operative values and dividing this outcome by the pooled standard deviation for loading rate and time-to-peak variables. Effect size (*d*) was interpreted as follows: *d* = 0.20 (small effect), *d* = 0.50 (medium effect), *d* = 0.80 (large effect), and *d* = 1.30 (very large effect).

## 3. Results

### 3.1. Patient-Reported Outcomes and Post-Operative Status

At post-operative measurements, all 10 patients reported a decrease of pain, according to the VAS scores, with an average decrease of 4.7/10. The FFI scores decreased as well, with an average of 32.2 points. Patients scored better on the SF-36 questionnaire, with an average increase of 16.7 points (Table 2). The post-operative radiographic analyses indicated a full union consolidation in eight patients and a partial-union consolidation in two patients. One of the latter patients received an additional 5 months of post-operative weight-bearing immobilization by means of a lace-up boot.

### 3.2. Kinematic Comparison

A waveform comparison of kinematic data is displayed in Figure 1. In general, all waveform patterns looked very similar between the pre-and post-operative data. The sagittal plane dorsiflexion ankle joint angle increased slightly, whereas the Chopart joint dorsiflexion diminished slightly. We observed no significant differences concerning the absolute range of joint motion in any of the foot joints (Figure 1 and Figure 3).

### 3.3. Kinetic Comparison

A waveform comparison of kinetic data is displayed in Figure 2. In general, we again observed very similar waveform patterns between pre-and post-operative data, except for the ankle joint power, where we observed a lowered power absorption of around 40%–60% of stance phase in the pre-operative stage (Figure 2). Concerning peak kinetic variables, no significant differences were found in any of the foot joints or the ankle joint, except for the ankle joint peak power absorption, which increased to 0.27 W/kg ± 0.12 W/kg in the post-operative condition, compared to 0.06 W/kg ± 0.03 W/kg (*p* = 0.001) pre-operation (Figure 3).

### 3.4. Biomechanical Load Comparison

A waveform comparison of the biomechanical load is displayed in Figure 4. The greatest difference was observed at initial contact, during which the pre-operative data showed no initial peak of biomechanical ankle load compared to the post-operative data. During pre-operative walking, patients furthermore showed a more gradual increase of ankle load, rather than the anticipated fast increase as was seen in the control and post-operative data. During the same phase of stance, pre-operative walking was characterized with a more dorsi-flexed position of the ankle compared to post-operative walking (Figure 4).

In Table 3, we calculated the amount of loading rate (BW/s) and time to peak from the waveform data mentioned above. From initial contact to the first loading peak, it took patients pre-operatively 0.17 ± 0.03 s, whereas post-operatively, this time to peak decreased to 0.15 ± 0.03 s (*p* = 0.009/moderate effect size of *d* = 0.61). The loading rate, calculated from the trough (lowest point between two peaks on a waveform) to the second peak, increased from 1.00 ± 0.64 BW/s pre-operatively to 1.57 ± 0.98 BW/s (*p* = 0.016/moderate effect size of *d* = 0.70) (Table 3).

## 4. Discussion

Arthrodesis of the tibiotalar joint is a widely used treatment approach in patients with isolated end-stage ankle OA. A plethora of studies demonstrated the efficacy of such an arthrodesis, with results suggesting significant post-operative pain relief and improved clinical outcome [3,4,7,23,24,25]. Despite this, patients still fear a loss of functionality after an ankle arthrodesis. This study used a recent development in gait analysis research that made it possible to quantify pre-and post-operative joint function by measuring kinematic and kinetic parameters of the ankle, Chopart, Lisfranc, and MTP 1 joints during walking.

The findings in this study revealed an interesting biomechanical phenomenon, in which patients with end-stage tibiotalar OA demonstrated a walking pattern that may originate from pain-mediated stiffness and inherent biomechanical deficiencies, as if they already possessed a functional arthrodesis (e.g., pre-operative ankle power absorption of 0.06 W/kg compared to the expected 0.52 W/kg in healthy controls). It can therefore be reasoned that the primary purpose of an ankle arthrodesis is to relieve pain in the tibiotalar joint without compromising the pre-operative gait biomechanics. As the tibiotalar joint is fused, one would expect to find a significantly stiffer tibiotalar joint during walking. Some background must be outlined before interpreting this finding. Due to methodological limitations of the used multi-segment foot model in this study, we quantified the walking biomechanics of the complete rearfoot, being a combination of the tibiotalar and subtalar joint [15]. This way, the found “rearfoot” range of motion of 9.7° (Figure 3) might potentially be solely attributed to the subtalar joint, therefore reducing the contribution of the tibiotalar joint to the total range of motion to near zero. The latter can be substantiated by a study on invasive in vivo measurements of rearfoot motion, which demonstrated that the subtalar joint carries a potential of on average 8.8° of sagittal plane range of motion during walking [26]. From a clinical point of view, this could be interpreted as that by fusing the tibiotalar joint, one does not alter the function of the rearfoot, as the subtalar joints remains similarly functional compared to their pre-operative status.

Kinetic parameters on top of kinematic parameters provide a significant added value in interpreting the joint function during walking. In this study, we found that the ankle joint kinetics are quite comparable between the pre-and post-operative measurements, except for the power absorption in the ankle joint. Joint power is calculated by multiplying the joint internal moment with the joint angular velocity. Since the ankle joint angular velocity only changed minimally, the greatest factor contributing to the increased power absorption must be the increased ankle joint internal moment. This might clinically suggest that patients were more tolerant towards rotational forces over the ankle joint during walking after surgery. A similar increase of mean ankle joint internal moment was found in the study by Brodsky et al., where they found a mean post-operative ankle moment increase of 0.2 Nm/kg [9].

A novelty in this study was that we, for the first time, quantified both the kinematics and kinetics of multiple foot joints during walking in patients undergoing an ankle arthrodesis. This granted us the opportunity to investigate the impact of an ankle arthrodesis on the biomechanical functioning of the Chopart, Lisfranc, and MTP 1 joints. The results of this analysis suggest that the kinetic behavior of the adjacent foot joints do not significantly differ post-operation, and that these therefore will not have to compensate for the potential loss of ankle joint function during walking, contrary to our hypothesis. It could therefore be clinically reasoned that the pre-operative acquired biomechanical patterns are still present after ankle OA, and that the loss of ankle joint power generation and absorption might lead to an adaptive strategy in these patients, during which they rely more on the passive supportive entities in the ankle and foot complex (ligaments, foot morphology, etc.). These might therefore be exposed to overuse, and gradually contribute to foot deformities. This consideration is preliminary, as we currently have not had the opportunity to compare these findings with other literature. Gait analysis is furthermore an indirect way to assess joint stresses during walking, and we therefore need to be cautious in interpreting these findings. On top of that, we quantified the distal foot joint biomechanics after a follow-up of one year, and these findings might differ when measured after a longer follow-up period. It is, however, of clinical interest to point out that the hypo-mobility of the distal foot joints, which might have originated from a pre-operative disuse, is still present after a one year follow-up. This might serve as an indication to initiate post-operative therapeutic mobilization exercises of these joints early on, in order to achieve a “more functional” foot.

Although we also included gait analysis data from asymptomatic control subjects, it was not our principal aim to compare patient data with these reference data. We merely included these data to provide some reference on what was expected to be considered as “normal values”. Overall, both pre-and post-operative gait analysis data in patients with isolated end-stage ankle OA was found to be significantly lowered compared to these reference data. We furthermore did not find that post-operative data resembled a return to normal gait, unlike earlier research in this patient population [6,24].

Concerning the ankle load, patients experienced an increase of load at initial contact compared to their pre-operative data. It can be reasoned that this is a consequence of the subjective post-operative pain relief, and that patients were therefore not afraid to properly land on their ankle joint during walking. It was even found that the post-operative ankle load at initial contact was similar to the control data. From initial contact to the first ankle load peak (around 20% of stance phase), the loading rate remained similar when compared pre-and post-operation. Since loading rate is the amount of force one endures over a certain amount of time, and we found that, pre-operatively, patients significantly elongated the time to peak, it could be suggested that the absolute amount of force endured was greater post-operatively. This might also be a positive effect regarding the pain relief. The loading rate from trough to second peak was increased post-operatively. This increase is due to the fact that post-operatively, patients experienced a lower trough in their loading waveform (Figure 4). Clinically, this means that patients, post-operatively, are more tolerant towards a rapid shift of joint forces during walking. Therefore, concerning the ankle load findings, ankle arthrodesis had a positive effect.

A first limitation was the inability to separately segment the subtalar joint, due to the current limitations of skin marker-based multi-segment foot models. Other foot models capable of measuring subtalar joint biomechanics during walking, such as an in vivo, bone-pin based model, were unethical to include in this patient population. Data of another study, using the latter model in healthy subjects, formed a basis to better interpret our current results obtained with the skin marker-based, multi-segment foot model [26]. A second limitation to this study was the relatively small patient population. Due to insufficient information in the literature, it was difficult to estimate the required sample size for sufficient study power a priori. We therefore opted to include a convenience sample size based on the number of participants in similar gait analysis studies. For this study, we included patients from a prospectively collected database consisting of around 200 patients with a rearfoot pathology; however, only 10 of these patients met the strict inclusion and exclusion criteria. Future research could perhaps broaden the inclusion criteria to provide more generalizable clinical significance for this quite heterogeneous patient population. Also, we studied this population at the one year follow-up, whereas future research should investigate the same outcome parameters after a longer follow-up period (e.g., five years), as these patients will return to more active lifestyles and increased participation, resulting in a more extensive use of the ankle and foot joints. This could then, perhaps even negatively, alter the current observed ankle and foot biomechanics. A third limitation was the fact that only barefoot walking data was collected, although in real life these patients often prefer shod walking. We opted not to include shod walking analyses in this study, due to the fact that multi-segment kinetic foot modelling is not yet considered accurate in shod conditions. Also, no frontal or transverse plane kinetic data was included. We did so to keep the study from becoming too comprehensive, and to keep the interpretations understandable. Frontal and transverse plane kinetics furthermore represent only a limited amount of the overall biomechanical functioning in the ankle and foot complex compared to sagittal plane kinetics, as studied in earlier research [13,27]. A last limitation might be the observed increase in walking speed after ankle arthrodesis in the patient group, as this could potentially effect some of the kinetic outcomes. However, a recent study suggested that these kinetic parameters significantly differ when walking speed increases by over 0.4 m/s (or 1.4 km/h) [28], whereas we currently only observed an increase of 0.09 m/s (or 0.3 km/h) between pre-and post-operative measurements. We therefore decided not to include walking speed as a covariate.

To conclude, patients with end-stage tibiotalar OA demonstrated a pre-operative biomechanical phenomenon during walking, in which pain-controlled tibiotalar stiffness represented that of a one year, post-operative, fused tibiotalar joint. Yet the pain relief achieved by tibiotalar arthrodesis in these patients did improve the loading patterns during walking. For the first time, the effect of an ankle arthrodesis on the kinematics and kinetics of multiple foot joints was investigated, with findings suggesting that these joints do not need to compensate for any function loss post-operation. From a clinical point of view, the results in this study suggest that patients with isolated end-stage ankle OA who are planned for an ankle arthrodesis do not need to fear a loss of function in their ankle and foot joints.

## Figures and Tables

**Figure 1 jcm-09-02594-f001:**
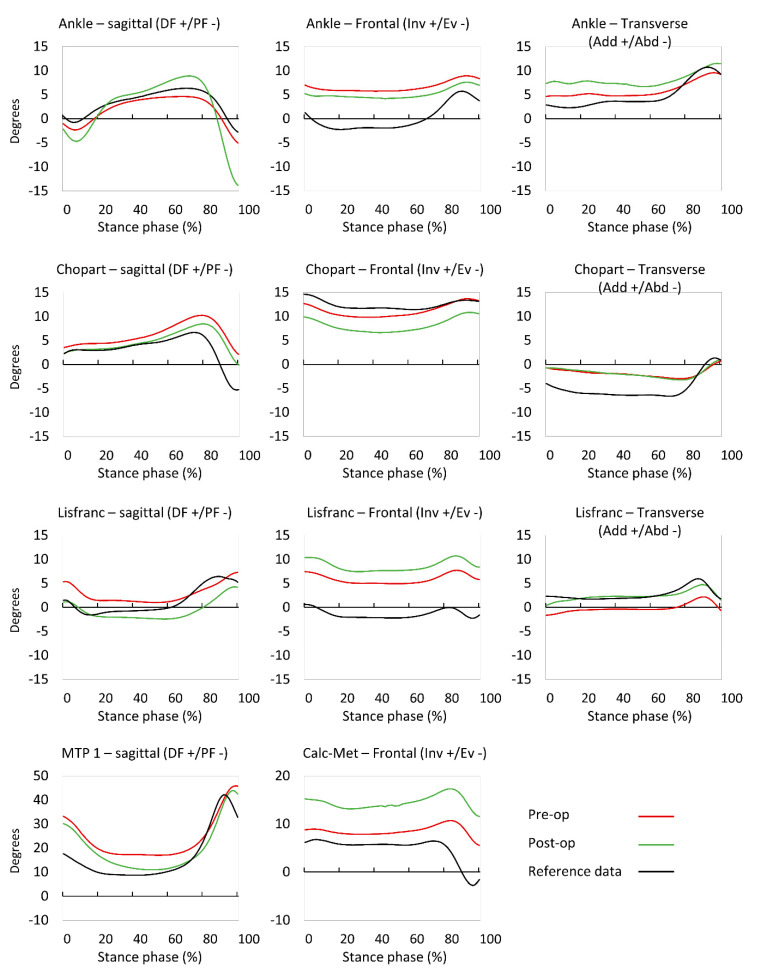
Control (dot/dashed line), pre-operative (dashed line), and post-operative (full line) kinematic waveforms of the ankle (**upper**) joint, as well as the Chopart (second row), Lisfranc (third row), and MTP 1 (**lower**) joints. PF: plantar flexion; DF: dorsal flexion; Inv: inversion; Ev: eversion.

**Figure 2 jcm-09-02594-f002:**
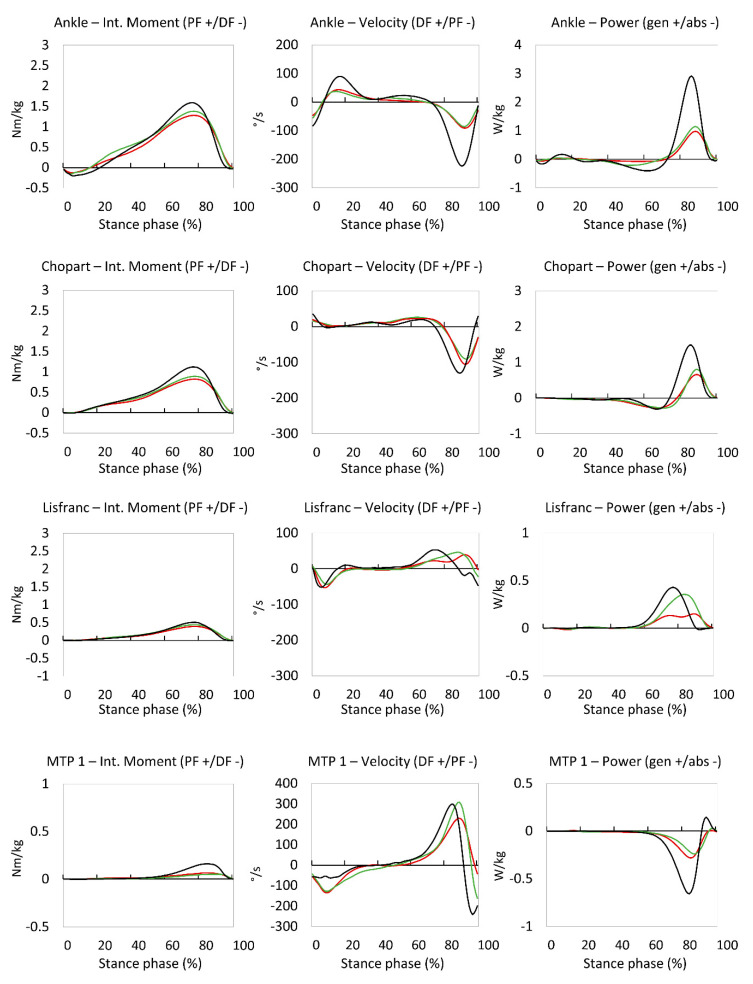
Control (dot/dashed line), pre-operative (dashed line), and post-operative (full line) kinetic waveforms of the ankle joint (**upper**), as well as the Chopart (second row), Lisfranc (third row), and MTP 1 (**lower**) joints. PF: plantar flexion; DF: dorsal flexion; gen: generation; abs: absorption.

**Figure 3 jcm-09-02594-f003:**
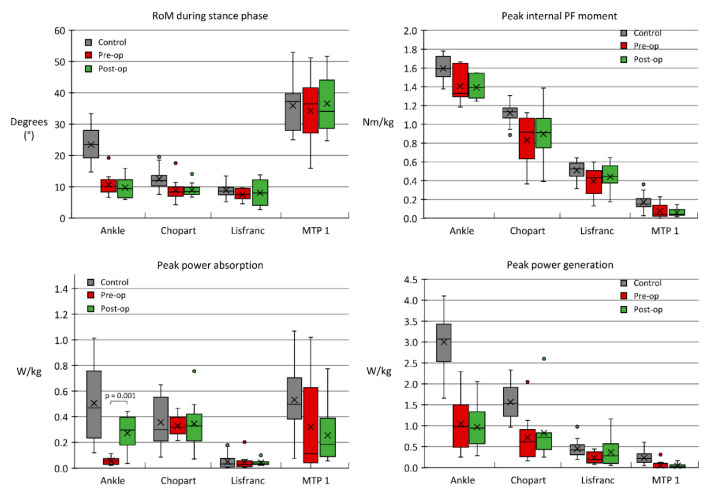
Distribution, mean (X) and median (horizontal line in box) of peak kinematic and kinetic variables during the stance phase of barefoot walking in control subjects (white), pre-operative (dark grey) subjects, and post-operative subjects (light grey). A paired *t*-test (α = 0.05) was performed to calculate differences between pre-and post-operative data. PF: plantar flexion.

**Figure 4 jcm-09-02594-f004:**
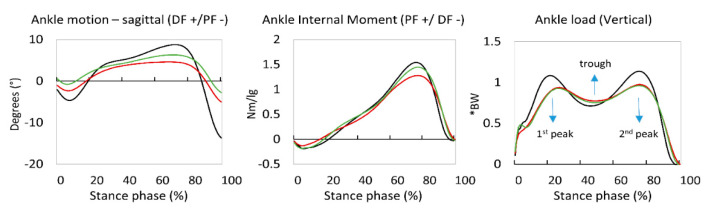
Waveform comparison of ankle motion (upper), ankle internal moment (middle), and ankle load (lower) of the control group (dot/dashed line), as well as and pre-operative (dashed line) and post-operative data (full line). Stance phase was time-normalized to 100%. DF: dorsiflexion; PF: plantar flexion; BW: bodyweight; Nm: Newton meter.

**Table 1 jcm-09-02594-t001:** Demographic and spatiotemporal data (median (range)).

Variable	Control (Reference Data)	Patients		
		Pre-op	Post-op	Paired *t*-test (*α* = 0.05)
Number of subjects (*n*)	17	10	10	
Side (L/R)	8/9	1/9	1/9	
Age (years)	36 (21–49)	53 (29–80)	54 (31–82)	
Body mass (kg)	72.7 (60.8–87.0)	76.5 (58.1–128.5)	80 (59.6–127.5)	
Height (cm)	178.1 (170.0–193.3)	168.5 (156.1–194.7)	168.1 (156.5–194.7)	
BMI (kg/m^2^)	22.5 (19.2–26.4)	25.5 (20.8–45.5)	25.7 (21.8–45.2)	
Walking speed (m/s)	1.32 (1.09–1.61)	1.00 (0.65–1.42)	1.09 (0.78–1.56)	0.002
Step time (s)	0.54 (0.48–0.58)	0.59 (0.49–0.74)	0.55 (0.47–0.62)	0.053
Cadence (steps/min)	110.8 (102.6–120.0)	101.1 (76.5–122.8)	106.9 (97.2–126.4)	0.090

BMI: Body Mass Index.

**Table 2 jcm-09-02594-t002:** Questionnaire and post-operative radiographic and clinical data.

	VAS Pain (/10)		FFI Score		SF-36		RX Status Consolidation	Complications	Post-Op Period without Immobilization
	Pre-op	Post-op	Pre-op	Post-op	Pre-op	Post-op			
Patient 1	7.5	2.5	61	29	59.5	79.3	Full union	No	14 months
Patient 2	3.0	1.5	39	29	51.9	68.8	Partial union	Delayed consolidation	11 ^1/2^ months
Patient 3	6.5	0.5	50	4	68.9	85.6	Full union	No	10 ^1/2^ months
Patient 4	4.5	0.5	36	7	57.3	70.7	Full union	No	8 months
Patient 5	8.5	0.0	50	0	68.0	88.6	Full union	No	9 months
Patient 6	3.5	0.5	46	4	87.3	90.6	Full union	No	9 months
Patient 7	7.5	1.0	39	14	56.2	78.6	Full union	Delayed callus formation	4 months
Patient 8	6.5	1.0	43	11	47.5	64.2	Partial union	Varus deformity	10 months
Patient 9	7.0	3.5	57	18	66.7	84.8	Full union	No	9 months
Patient 10	5.0	1.5	21	4	68.9	87.9	Full union	No	19 months

VAS: Visual Analog Scale; FFI: Foot Function Index; SF-36: Short-Form 36; RX: radiography.

**Table 3 jcm-09-02594-t003:** Loading rate (BW/s) and time to peak (s) during stance phase.

Variable	Control Group (*n* = 17)	Patients (*n* = 10)			
		Pre-op	Post-op	*p*-Value (*α* = 0.05)	Effect size (*d*)
LR (BW/s): IC to first peak	8.58 ± 1.17	5.34 ± 1.61	5.64 ± 1.81	0.353	0.17
Step time (s): IC to first peak	0.11 ± 0.01	0.17 ± 0.03	0.15 ± 0.03	0.009	0.61
LR (BW/s): trough to second peak	2.77 ± 0.92	1.00 ± 0.64	1.57 ± 0.98	0.016	0.70
Step time (s): trough to second peak	0.16 ± 0.02	0.15 ± 0.02	0.14 ± 0.02	0.411	0.21

LR: loading rate; BW: bodyweight; IC: initial contact.

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
