# Peer review of "The Biomechanical Behavior of Distal Foot Joints in Patients with Isolated, End-Stage Tibiotalar Osteoarthritis Is Not Altered Following Tibiotalar Fusion"

_jcm, 2020, doi:10.3390/jcm9082594_

Round 1

Reviewer 1 Report

The authors made good effort to improve the manuscript. My personal questions and recommendations to the authors were answered appropriately, and in my opinion the personally suggested modifications are well addressed in this revised version. 

Author Response

We would like to sincerely thank the reviewer for his kind words and much appreciated feedback on the manuscript.

Reviewer 2 Report

Article is presenting a lot of data which should be semplified in their graphical expression, presenting to date the risk to be confunding

Author Response

We would like to sincerely thank the reviewer for their time and efforts to review our study.

In accordance with Reviewer 1, we made significant changes to the figures to simplify and improve the understandability. 

Reviewer 3 Report

Well-done study, and nicely presented results! However, I have some remarks:

  • inclusion criteria: out of 200 cases, just 10 patients have met the inclusion criteria, mainly of having an isolated tibiotalar OA. The preoperative pain score was quite low (average 4.7 [3.0 - 8.5]), indicating that the used ankles were in a rather good shape, i.e. ideal for getting a successful result with ankle fusion.
  • Post-op period without immobilization was only 10.5 [4 - 19] mo, meaning the full return in a normal life was not achieved yet. Usually, 1 to 2 years are necessary to get a patient after such a major surgery back to normal life, meaning in a status where the patient is back for using his/her ankle again normally. The fact that obtained results from this study were good-excellent might be a result of patient's selection; thus it must be questioned whether the result would be similar in a patients with an usual ankle OA, especially posttraumatic ankle (which accounts to 60 to 80% of end-stage OA ankles according to the current literature)
  • It is suggested that patient's decision to go for an ankle fusion was mainly based on pain with subsequent reduction of life quality; as the ankle became fused, patient's expectations were fulfilled resulting in a high satisfaction at this early stage. With more FU time, however, patients will become more demanding and they will use their ankle more extensively which may result in limitations and eventually even pain, as reported by the few long-term FU studies after ankle fusion, indicating that an assessment before 7 to 10 years after surgery might be critical.
  • As mentioned by the authors, one of the most conflicting problems in 3D-gait analysis is the reliability of the markers. As using pins is not acceptable from ethical standpoint, the markers are fixed to the skin, instead. I do not know the performance of the used software / foot model in detail; however, I assume that a major issue in the present study was the fixation of the markers to the skin in a reliable and reproduceable way. My question is whether the authors have made test-retest investigation to determine the reliability of the technique used. In any case, I definitively feel that this aspect must be discussed, and it might be also included in the limitation section.

I definitively feel that the discussion section must be more critical, and I would suggest adding clinical implications where these aspects are discussed. In particular, the limitations should be more specific and address also these concerns.

Nevertheless, this is a well-done study with some new and exciting results that are worthwhile to be shared with the community.

Author Response

This manuscript is a resubmission of an earlier submission. The following is a list of the peer review reports and author responses from that submission.

Round 1

Reviewer 1 Report

JCM-812456

Title: The biomechanical behavior of distal foot joints in patients with isolated end-stage tibiotalar osteoarthritis is not affected following tibiotalar fusion

Journal: Journal of Clinical Medicine

General impression:

In this interesting study, the biomechanical behavior of multiple joints (ankle, Chopart, Lisfranc, MTP1) was investigated in (n=10) patients with tibiotalar arthrodesis and (n=17) healthy controls. The manuscript is well written and concise. Results and conclusions are supported by the data, figures and tables. Novelty and limitations are clearly described.

Comments and suggestions:

  • What is the repeatability of the measurements and analyses in healthy controls? I’m missing a second time point in the healthy control group.

  • In the patient group, one could imagine that patients may react differently after treatment during certain walking, step time and cadence tests and try to move faster for example. Could you elaborate in more detail on possible bias?

  • Please elaborate on the exclusion criteria regarding medical conditions that may affect normal gait, such as…

  • The age, body mass, height and BMI were not matched with the patient groups. Could you please elaborate on the possible effect on the outcome?

  • In the introduction a study of Brodsky (2016) was mentioned, stating that the ankle joint ROM increased after tibiotalar fusion. Please elaborate on the rational in more detail clarify.

  • You hypothesis can me improved, since you hypothesize that the kinematic behavior will significantly ‘differ’, where ‘impaired’ may be more sufficient.

  • The limitations section should be moved to the last paragraph of the discussion, before the conclusion.

  • The figures/graphs can me improved since they are difficult to comprehend due to the small size and missing lines/axes.

Reviewer 2 Report

The aim of this study was to “investigate the post-operative effect of an arthrodesis on the ankle and foot joint biomechanics” (21-22).

One of the main compensatory mechanisms following ankle arthrodesis is altered kinetic and kinematics at the subtalar joint. However, the Biomechanics of this joint was not investigated in this study due to access limitations by conventional Gait Analysis techniques.

Before applying conventional gait analysis using external passive markers to a functional activity such as locomotion, the authors had to demonstrate their goal in isolated controlled conditions. They confuse terms such as ROM which are the limits of motion of a joint with the motion at the joint during locomotion. This confuses the reader to believe the absurd notion that fusing the ankle does not limit its range of motion!! If no motion occurred at the ankle pre-operatively due to guarding and no motion occurs post-operatively due to fusion you cannot reach the absurd conclusion that arthrodesis had no effect.

With what accuracy and resolution can a conventional gait analysis system detect subtle changes occurring in the midfoot and forefoot of a subject during locomotion? This is a major problem in this study particularly as the motion at these joints is extremely small and within the accuracy and resolution limits of the measuring system itself.

The patient population is much too small for statistical significance when the variations are known to be huge no only inter-subject but intra-subject as well.

Limiting the study to 2D vs 3D is another serious problem in this study. The authors justify it based on not to produce and report complex results difficult to interpretation. However, the 3D analysis could easily effect the 2D data representation so there is no justification in a-priori doing a 2D analysis.

Line 215-271: But the pre-operative gait is already compromised due to the presence of arthritis. The purpose of the treatment is not just to reduce pain but also to improve function!

Line 223-227: Another false argument. The fact that the Subtlar joint has a range of 8.8 degrees does not mean that during normal walking all this range is being used and therefore the claim that the arthrodesis does not affect subtalar biomechanics is clearly unjustified.

Line 238-239: If this is indeed the novelty of this study, the accuracy and resolution of the system must have been carefully examined and reported. No such reporting was provided here.